# Stereotactic Body Radiotherapy as a Curative Treatment for De Novo Mucosal Carcinoma of the Head and Neck: A Feasible Alternative Option for Fragile Patients with Small Lesion: A Systematic Review [note 1]

**DOI:** 10.3390/cancers16112096

**Published:** 2024-05-31

**Authors:** Primož Strojan, Marko Kokalj, Gaber Plavc, Sweet Ping Ng, Sandra Nuyts, Carlos M. Chiesa-Estomba, Avraham Eisbruch, Remco de Bree, James C. H. Chow, Antti A. Mäkitie, Fernando Lopez, Nabil F. Saba, Alfio Ferlito

**Affiliations:** 1Department of Radiation Oncology, Institute of Oncology Ljubljana, 1000 Ljubljana, Slovenia; mkokalj@onko-i.si (M.K.); gplavc@onko-i.si (G.P.); 2Faculty of Medicine, University of Ljubljana, 1000 Ljubljana, Slovenia; 3Department of Radiation Oncology, Austin Health, The University of Melbourne, Melbourne, VIC 3000, Australia; sweetping.ng@austin.org.au; 4Laboratory of Experimental Radiotherapy, Department of Oncology, Leuven Cancer Institute, University Hospitals Leuven, 3000 Leuven, Belgium; sandra.nuyts@uzleuven.be; 5Department of Otorhinolaryngology-Head & Neck Surgery, Hospital Universitario Donostia, 20014 San Sebastian, Spain; carlosmiguel.chiesaestomba@osakidetza.eus; 6Biodonostia Research Institute, Faculty of Medicine, Deusto University, 20014 San Sebastian, Spain; 7Department of Radiation Oncology, University of Michigan Medicine, Ann Arbor, MI 48109, USA; eisbruch@med.umich.edu; 8Department of Head and Neck Surgical Oncology, University Medical Center Utrecht, Utrecht University, 3584 CX Utrecht, The Netherlands; r.debree@umcutrecht.nl; 9Department of Clinical Oncology, Queen Elizabeth Hospital, Hong Kong Special Administrative Region, Hong Kong, China; cch932@ha.org.hk; 10Department of Otorhinolaryngology-Head and Neck Surgery, Research Program in Systems Oncology, University of Helsinki and Helsinki University Hospital, FI-00014 Helsinki, Finland; antti.makitie@helsinki.fi; 11Department of Otorhinolaryngology-Head and Neck Surgery, Hospital Universitario Central de Asturias (HUCA), 33011 Oviedo, Spain; lopezafernando@uniovi.es; 12Instituto Universitario de Oncología del Principado de Asturias, Instituto de Investigación Sanitaria del Principado de Asturias, University of Oviedo, 33011 Oviedo, Spain; 13Department of Hematology and Medical Oncology, Winship Cancer Institute, Emory University School of Medicine, Atlanta, GA 30322, USA; nfsaba@emory.edu; 14Coordinator of the International Head and Neck Scientific Group, 35125 Padua, Italy; profalfioferlito@gmail.com

**Keywords:** stereotactic body radiotherapy, head and neck cancer, mucosal tumor, toxicity, local control, survival

## Abstract

**Simple Summary:**

Stereotactic body radiotherapy (SBRT) is a highly precise, non-invasive, and image-guided form of hypofractionated radiotherapy aimed at well-defined small targets. It is characterized by superior conformity of dose distribution, steep dose drop-off gradients, and short overall treatment time, which allow effective protection of surrounding normal tissue from high radiation doses, potentially resulting in lower toxicity and a more potent biological effect. Currently, SBRT is an established curative therapy for certain cancers and some benign tumors, as well as a valuable palliative option. There is limited experience with SBRT for de novo mucosal carcinoma of the head and neck. A systematic review of the literature and subsequent analysis of 124 patients from five studies treated between 2012 and 2020 confirmed that SBRT is an effective and relatively safe treatment for small tumor targets in de novo mucosal carcinoma of the head and neck. Standardization of SBRT practice and well-designed prospective clinical trials are highly needed.

**Abstract:**

Stereotactic body radiotherapy (SBRT) is characterized by a high dose per fraction, well-defined small targets, superior dose conformity, and a steep off-target dose gradient. A literature search was conducted to examine the experience with SBRT as a curative treatment for newly diagnosed mucosal carcinoma of the head and neck (MCHN). Four retrospective case series and one prospective phase I clinical trial published between 2012 and 2020 described 124 patients. SBRT was mainly performed in older patients with different tumor sites. The median size of the planning target volumes ranged from 5.3 to 41 cm^3^. Different approaches were used to create margins. In two studies, limited elective nodal irradiation was performed. The equivalent doses used were 60–83.33 Gy delivered in five fractions. Considerable heterogeneity was observed in the radiation dose specification. The incidence of grade ≥3 late toxicity was 0–8.3%, with local and regional control ranging from 73% to 100%. Improved or stable quality of life after SBRT was reported in two studies. Curative-intent SBRT for de novo MCHN appears to be an effective and relatively safe treatment for small tumor targets, preferably without concomitant elective tissue irradiation. Standardization of SBRT practice and well-designed prospective clinical trials are needed to better define the role of SBRT in this setting.

## 1. Introduction

Stereotactic body radiotherapy (SBRT) represents a new paradigm in the arsenal of radiotherapy techniques [1]. The radiobiological rationale for SBRT lies in the delivery of a few fractions of a high dose within a relatively short overall treatment time, resulting in a more potent biological effect [2]. Its advancement is primarily due to technological advances, particularly in imaging, radiation delivery systems, planning, and immobilization [3]. SBRT is characterized by the use of high daily doses of radiation, typically delivered in 1–5 fractions (hypofractionation) with a high level of precision to a well-defined and small target volume. With superior conformity of the dose distribution and steep dose drop-off gradients, SBRT enables effective protection of the surrounding normal tissues, potentially translating to lower toxicity and a better quality of life in the long term. Due to its non-invasive nature and shorter treatment duration, it proves to be more time- and potentially cost-efficient than conventional radiotherapy. This allows patients to complete their treatment quickly and potentially resume their normal home or work routine [4].

SBRT is already an established therapeutic option for the treatment of certain cancers and some benign tumors [3]. It is an established curative treatment option for early-stage non-small cell lung cancer [5], primary liver tumors [6], and localized prostate cancer [7]. In addition, SBRT is effectively used to alleviate symptoms caused by metastases in different parts of the body, especially in the context of oligometastatic disease, possibly in combination with immunotherapy [8,9]. Proper patient selection with focused and precise targeting of the tumor potentially provides favorable quality of life results despite ablative radiation doses [10,11].

Cancers of the head and neck account for 4.5% of all cancer diagnoses worldwide and 4.6% of all cancer deaths [12]. The majority are squamous cell carcinomas originating from the mucosa of the upper respiratory and digestive tract structures. The therapeutic scenarios used in the treatment of squamous cell carcinoma of the head and neck were developed more than 20 years ago, with fractionated radiotherapy, 1.8–2.2 Gy daily fractions over 6–7 weeks with concurrent systemic therapy still being the basis for the non-surgical curative treatment [13]. The role and potential of SBRT in de novo mucosal carcinoma of the head and neck (MCHN) is poorly defined, either as the mainstay treatment (with/without systemic therapy), as a boost before or after surgery, or as a boost after a course of conventionally fractionated radiotherapy. More data are available on the use of SBRT for re-irradiation and oligometastatic disease [14,15]. However, in these cases, the efficacy and role of SBRT are limited due to the inherent limitations of re-irradiation in terms of tissue radiation tolerance and/or systemic spread of disease.

The aim of this systematic review is to provide a summary of the current literature on the use of SBRT as a primary treatment modality for de novo MCHN with a potentially ablative dose and the future prospects for SBRT in this setting.

## 2. Materials and Methods

A systematic review of the existing literature was conducted according to the PRISMA guidelines using the PICOS criteria (Table 1) [16] and was registered with the Research Registry under the identification number: reviewregistry1826. The study focused on patients with treatment-naive MCHN who received potentially ablative radiotherapy doses using the stereotactic technique. Additional inclusion criteria were that dose delivery had to be hypofractionated with 5 or fewer fractions, an equivalent dose in 2 Gy fractions (α/ß = 2, EQD2_10_) ≥60 Gy was administered in an SBRT-like fashion, and the report was written in English.

All published full-text articles listed in the PubMed (MEDLINE), Web of Science, and Scopus databases from inception to 30 October 2023 were included in the study. Search terms included radiosurgery, stereotactic radiosurgery, stereotactic body radiotherapy, stereotactic ablative body radiotherapy, stereotactic radiotherapy, stereotactic radiation, stereotactic radiation therapy, CyberKnife radiosurgery, Gamma Knife radiosurgery, linac radiosurgery, linear accelerator radiosurgery AND head and neck neoplasms, head and neck cancer, head and neck tumor, cancer of head and neck, head neoplasm, head cancer, neck neoplasm, neck cancer, cancer of the head, or cancer of the neck. In addition, the publication reference lists cited in the articles found were checked.

The results were screened in three steps. In the first two steps, the selection was based on a review of the titles and abstracts of the publications. First, duplicates, non-English texts, animal studies, and texts that did not meet the characteristics of a full-text article (abstracts and other reports from congresses and professional/scientific meetings, non-peer-reviewed correspondence, letters to the editor and replies, editorials, case reports/series with up to three cases described, review articles) were excluded. In the second step, records were excluded if they did not contain SBRT, referred to the treatment of other tumor types (histology other than squamous cell carcinoma, other body sites), addressed re-irradiation or metastatic disease, did not report treatment outcomes (but focused on tissue tolerance to radiation, side effects, technology, etc.), or did not allow data extraction for the subgroup of MCHN.

In the third step, a detailed review of the remaining publications was performed. After reviewing the full texts, those that met the inclusion criteria were selected. The following data were extracted from the eligible studies: type of study (retrospective, prospective case series, randomized clinical trial), number and age of patients, tumor location, histology and stage, SBRT treatment details (delivery platform, planning target volume [PTV] size, elective nodal irradiation, dose prescription data, biological effective dose [BED], EQD2, image guidance), use of systemic therapy and toxicity details (acute, late grade 3 or higher), treatment outcome (local and regional control), patient survival (progression-free survival and overall survival), and quality of life assessments results. The risk of bias of the included studies was assessed independently by 2 of 3 authors (PS, MK, GP) using the ROBINS-I tool for non-randomized studies, which classifies studies into those at low, moderate, serious, and critical risk of bias [17]. Disagreements were discussed until consensus was reached.

Due to differences in fractionation regimens and total doses delivered, physical dose data were converted to BED using the formula:BED_α/β_ = nd(1 + d/[α/β]),
where n is the number of fractions; d is the dose per fraction; and α and β are the coefficients of dose and the dose squared in the linear-quadratic model, respectively. For tumors, α/β = 10 Gy was used [18].

The conversion to the equivalent dose delivered in 2 Gy fractions (EQD2) was calculated using the following formula:EQD_α/β_ = nd(d + α/β)/(2 Gy + α/β).

The primary objectives of the analysis were SBRT toxicity and local control; secondary objectives were regional control, progression-free survival (PFS), overall survival (OS) and quality of life.

## 3. Results

After reviewing the databases, 985 data records were identified (Figure 1). Based on the previously defined criteria, 945 records were excluded in the first two steps and 40 records were examined in the third step. Of these, 17 were original studies in which SBRT was used either as a boost after previous conventionally fractionated RT (N = 12), or before (neoadjuvant, N = 4) or after surgery (postoperative, N = 1). Finally, only five records, published between 2012 and 2020, formed the basis for the following analyses [19,20,21,22,23].

### 3.1. Characteristics of the Eligible Studies

Four studies were retrospective reviews and one was a prospective phase I trial in which only the arm with the highest dose met the EQD2_10_ criterion of ≥60 Gy (Table 2). The included patients were treated between 2002 and 2018 (most of them after 2010), and only two studies treated more than 15 patients. There was one site-specific study (glottis) in which patients were about two decades younger (mean age 61 years) than in the other four studies, which treated mucosal tumors of different parts of the head and neck in patients with a median age of >80 years. In the two larger study groups, which together accounted for 75% of the cases, more than two-thirds of the tumors were histologically squamous cell carcinoma, while the exact stage of disease (T, N stage) was not specified or reported for the de novo MCHN subgroup [22,23]. For the majority of patients, the reason for referral to SBRT was comorbidities and consequent medical inoperability.

According to the ROBINS I tool for assessing the risk of bias in the included studies, the overall risk of bias was rated as severe for all four retrospective studies and moderate for the prospective study. The risk of bias in each domain is summarized in Table 3. As the retrospective studies are case series, an important source of bias can be found in all three domains.

### 3.2. Treatment Details

SBRT was performed using either CyberKnife or a gantry-based linear accelerator; both options were used in one study (Table 4). In three studies, the PTV was created with a 2–5 mm extension of the gross tumor volume (GTV); in one study the clinical tumor volume (CTV) was created beforehand and a margin of 3 mm was added to form the PTV; and in one study the formation of the PTV was not described. The median size of the treated PTV ranged from 32 to 41 cm^3^ in two studies, in the early glottic tumor group it was 5.3 cm^3^, while in another study the reported median size of the GTV was 33.2 cm^3^. In one study this information was not reported. The prescribed EQD2_10_ doses (α/β = 10 Gy) ranged from 60 to 83.33 Gy (<70 Gy in 4 of 5 studies) and were administered in 5 fractions, using a schedule of 1–3 fractions per week. The method of dose prescription varied widely between studies, from prescribing it to a specific isodose line to specifying the dose at 95% of GTV or PTV volume. In two larger studies, in a small (but unspecified) proportion of patients, selected adjacent lymph nodes were also electively irradiated with a lower dose. Prior to each fraction, daily image guidance was performed using CBCT and/or kV imaging to ensure appropriate and accurate patient positioning. In three studies, some patients received systemic therapy during SBRT [19,20,23].

### 3.3. Toxicity

No grade 3 or higher acute toxicity was observed in two studies, and in the others the cumulative incidence ranged from 3% to 52.1% (Table 5). In the latter case, grade 3 radiomucositis and radiodermatitis were observed in 32.5% and 19.6% of patients, respectively. Late toxicities of grade 3 or higher were reported in up to 8.3% of cases. Necrosis was observed most frequently (in three cases), while late mucositis and skin ulceration were observed in one case each. In one study, not a single case of late ≥G3 toxicity was reported.

### 3.4. Tumor Control and Survival

The duration of the mean/median follow-up (FUP) in the individual studies ranged from 10.5 months to 36 months and was ≥12 months in four out of five studies (Table 5). Local control was achieved in the majority of patients, ranging from 73% (at 1 year) to 100%. No nodal recurrences were observed in two studies, with a median FUP time of 20 and 20.5 months in which elective irradiation of the regional lymphatics was omitted [20,21]. In one study, regional control was 78.6% after a median 3-year FUP; in another, 73% after one year of FUP; and in the third study, locoregional control outside the PTV was 89% after one year [19,22,23]. OS varied between 40% and 100%.

### 3.5. Quality of Life

Quality of life was only assessed in two studies (Table 5). In the first, the University of Washington Quality of Life Revised survey was used for this purpose. In 7 (of 12) patients who completed questionnaires before and after SBRT, 71% had improved or stable quality of life compared to baseline after a median FUP of three months (range 0–15 months) [20]. The second study used the MD Anderson Dysphagia Inventory, which showed excellent swallowing scores initially and during the recovery period (up to 12 months after SBRT), and the Voice Handicap Index (VHI), which indicated marked voice impairment initially, followed by rapid and marked improvement over time [21].

## 4. Discussion

The main observation of our systematic review is that SBRT with an ablative EQD2_10_ dose of ≥60 Gy was rarely used as treatment for de novo MCHN. It was mainly used in older patients where the poorer health status justifies the omission of elective irradiation of larger tissue volumes around the gross tumor locally and/or regionally. In addition, considerable heterogeneity in SBRT practice was noted in the studies analyzed, particularly in terms of margins to account for microscopic disease, dose prescription, and fractionation pattern.

Only five studies that met the inclusion criteria were found in the literature. Even in these, only a subset of patients, 124 in total, were eligible for analysis [19,20,21,22,23]. Most were older with an average age of ≥80 years. One exception was the report by Sher et al., the only prospective and site-specific study that exclusively included patients with early-stage vocal cord cancer who were, on average, two decades younger [21]. For these reasons, this study must be considered separately.

The small number of studies identified and patients treated is not surprising given that existing therapeutic scenarios for non-surgical treatment of de novo MCHN involve elective treatment of large volumes of tissue to eliminate microscopic clusters of tumor cells likely to be present around the gross tumor and in the neck nodes. High daily doses, typical of SBRT, and a total EQD2_10_ dose of ≥60 Gy can only be used to irradiate well-defined smaller targets. For larger tumors, there is an increased likelihood of marginal failure due to the rapid and sharp dose fall-off at the edge of the high-dose volume, along with a higher likelihood of tumor cell spread to surrounding tissue, which may be missed or undertreated [4,24]. Furthermore, irradiating larger tissue volumes in this case would increase the risk of severe acute and, especially, late radiation damage, as late-responding tissue is more sensitive to a higher fraction dose [25]. In fact, dose delivery was highly focused in all five studies analyzed: no CTV was created around the GTVprimary or it was created to a limited extent only in selected cases (Table 4). The median volume of the irradiated PTVs ranged from 5.3 to 41 cm^3^, which corresponds to the volume of a sphere with a diameter of 21.6 or 42.8 mm. Elective irradiation of the neck, when indicated, included only a limited number of ipsilateral neck regions and lower daily doses (typically 5–6 Gy/fraction), which was found to be well tolerated and safe [26].

Under these conditions, the reported incidence of late toxicity of any grade ≥3 ranged from 0% to 8.3% (Table 5). Given the retrospective design of all studies except the study by Sher et al. (in which the irradiated PTVs were the smallest) [21], it could be assumed that the toxicity figures in the analyzed reports may have been underestimated and the incidence of serious adverse events may be higher. For example, Ozyigit et al. reported serious toxicities (brain necrosis) in 2 out of 6 patients (33%) with de novo tumors of the nasal cavity/paranasal sinus treated with primary SBRT, despite the low tumor EQD2_10_ of 49.58 Gy and 54.69 Gy (35 Gy and 37.6 Gy in five fractions) [27]. In this series, the dose was normalized to a 75% isodose line that included 95% volume border of the PTV (CTV + 1 mm; CTV = GTV + 5 mm), the size of which was not specified. In 182 patients with oropharyngeal carcinoma reported by Baker et al., the estimated cumulative incidence of grade ≥3 late toxicities at five years was 28% after a hypofractionated SBRT boost to the primary tumor of 3 × 5.5 Gy (EQD2_10_ 21.31 Gy) following conventional fractionated intensity modulated radiotherapy (IMRT) of 46 Gy to the macroscopic tumor and elective neck regions [28]. The most commonly observed toxicities were mucosal ulceration or soft tissue necrosis (including carotid blow-out and tracheal necrosis with fatal hemorrhage), dysphagia, and osteoradionecrosis, with an increased risk observed in smokers at diagnosis and in patients with Charlson Comorbidity Index ≥2. In this study, the elective phase CTV, which covered the GTVprimary with a 1 cm margin, was extended by 3 mm to form the PTV of SBRT; the SBRT dose was prescribed up to the 80% isodose line [28]. Further escalation of the tumor dose is expected to lead to an additional increase in the incidence of severe late toxicity [29,30].

The importance of limiting the irradiated volume in SBRT cannot be overestimated. In this context, reference should be made to a prospective clinical phase I trial by Sher et al. to evaluate the safety and efficacy of SBRT in early-stage glottic cancer [21]. By limiting the target volume to the primary tumor and with an EQD2_10_ of 65.72 Gy, the incidence of any grade 3 late toxicity was 8.3% (soft tissue and arytenoid necrosis required intensive care level hospitalization to heal in 1/12 of patients). On the other hand, in a similarly designed phase I study with a median follow-up of 14.5 months, Kang et al. reported two (33%) grade 3 laryngeal inflammation events (vocal cord ulcer and arytenoid cartilage necrosis requiring supraglottic laryngectomy) in six patients [31]. In this study, the EQD2_10_ for the PTVprimary (PTV = GTV + 3 mm) was 68.75 Gy (55 Gy in 11 fractions) and 46.47 Gy (40.7 Gy in 11 fractions) for the remaining larynx, administered every other day or twice a week. Consequently, the exclusion of elective larynx irradiation and the more intensive daily imaging protocol used in the study by Sher et al. had a dramatic impact on the results and conclusions of the two studies. Thus, SBRT for early-stage glottic cancer was declared feasible and safe in the first study, whereas the second study was terminated prematurely [32].

Even if there is no clear correlation between the duration of treatment and the occurrence of late toxicities, the detrimental influence of the significantly shorter overall duration of treatment with SBRT (2–2.5 weeks) compared to conventionally fractionated radiotherapy (6–7 weeks) cannot be completely ruled out [33]. For example, a longer persistence of mucosal ulcers was observed after a hypofractionated SBRT boost than after conventionally fractionated IMRT (total treatment duration shortened from 6–7 to 5 weeks), although patients in the former group were younger, in a better performance status, had smaller tumors, and were less likely to be treated with concurrent chemotherapy [34].

With highly focused irradiation of rather small targets, as described above, and a predominant EQD2_10_ of less than 70 Gy, the reported local control in the analyzed series was between 70% and 100% with a follow-up time of mostly less than 2 years (Table 5). Despite the absence or very limited elective irradiation of regional lymphatics, regional control was also above 70%, while OS figures mainly reflect the advanced age of patients treated with SBRT and cannot be used as a measure of treatment efficacy. However, the reported overall survival (OS) figures are comparable to those from series in which patients of similar age were treated with conventional fractionated radiotherapy, either as sole therapy or in combination with chemotherapy or surgery [35,36]. With a significantly shorter treatment duration, which is an important consideration in this age group of patients, and at least comparable, if not more favorable toxicity profile, SBRT appears to be an attractive option for this patient group. In this context, it is worth reiterating the study by Sher et al. in which SBRT of early-stage glottic tumors resulted in 100% local control after a median follow-up of 25.7 months, with voice and swallowing ability preserved during this period [21]. Vargo et al. also reported improved or stable quality of life scores after SBRT compared to baseline as assessed by the University of Washington Quality of Life Questionnaire [20].

Despite the relatively low EQD2_10_ used in the majority of SBRT patients compared to conventional treatment regimens, the local and regional disease control is quite encouraging, especially considering the advanced age, poor general condition, and short life expectancy of these SBRT patients. This may be due to the unique mechanisms of cell killing in SBRT that are not directly related to the four Rs of conventional fractionated radiotherapy, which determine its net effect on the tumor [2,37]. In addition to direct cell killing, the high dose per fraction deteriorates the intratumor microenvironment by destroying the tumor vasculature, resulting in indirect tumor cell death. Consequently, the destruction of tumor cells by SBRT leads to a massive release of tumor antigens, stimulation of anti-tumor immunity, and thus suppression of the regrowth of local or systemic tumor foci [37]. However, other authors claim that no other phenomena besides the classic four Rs are involved in the tumor response after SBRT, with the probable exception of enhanced tumor immunity induced by high doses of irradiation in some tumors [38]. Therefore, research into combinations of SBRT and systemic immunotherapy is highly warranted, especially considering the focused dose delivery of SBRT, and thus the need for effective eradication of microscopic tumor nests around the primary tumor and in the regional lymphatics. In addition, extensive irradiation of lymphatic stations commonly employed in the clinic can reduce the effect of concomitant immunotherapy. It is suggested that avoiding elective nodal irradiation may support the immunologic responses that may be enhanced by immunotherapy [39]. In the studies we analyzed, systemic therapy was heterogeneous and rarely used, so their results do not allow us to draw conclusions in this regard. However, in their analysis of the role of SBRT in combination with immunotherapy in the neoadjuvant setting, Mohamad et al. are optimistic, although the omission of elective nodal irradiation during neoadjuvant SBRT was found to increase the risk of regional nodal recurrence despite the use of immunotherapy, even in the presence of favorable human papillomavirus-related oropharyngeal cancer [40].

Our review is limited by caveats associated with each of the studies analyzed, ranging from the small number of cases included to incomplete data reporting, and the heterogeneity of the studies analyzed. Not only are all but one study retrospective, and the overall risk of bias was considered severe; even within individual studies, it can be a challenging to clearly separate those with de novo MCHN from other patients. A significant limitation resulting from the retrospective nature of most studies is also the lack of information on the TNM stage of the disease, which would allow a fair assessment and comparison of local control and survival outcomes between studies. In addition, the use of elective radiation volumes with the associated definition of volume margins and dose fractionation schemes varied considerably between studies. Another important difference in SBRT practice is the pattern of dose prescription. It varied from the standard mode, where the dose was prescribed on the 100% isodose line, to prescribing on a specific isodose line between 57% and 90%. In this case, the 100% isodose value determines the maximum dose, and there is a varying degree of dose heterogeneity within the PTV, which may significantly affect the prescribed dose information and could lead to significantly different biological doses. Considerable heterogeneity among experienced SBRT centers regarding patient selection, prescription, and treatment delivery approaches was already noted by Karam et al. in their review of SBRT practices for head and neck cancer [1]. However, this observation argues for standardization of dose prescription and reporting in SBRT for head and neck cancer to allow for more credible comparisons of treatment outcomes between different institutions [41].

## 5. Conclusions

SBRT as a curative treatment for de novo MCHN may be a viable and relatively safe treatment option when the target volume is small, preferably without additional elective irradiation of adjacent tissue. Our review found that it is primarily used in the treatment of elderly patients with poorer general health who are not candidates for long-course conventional fractionated radiotherapy or surgery. Nevertheless, this is a relatively rare indication and contrasts with re-irradiation for recurrent or new primary head and neck cancer in a previously irradiated volume, where the results of salvage SBRT in terms of tumor control and toxicity are very similar to the results of standard re-irradiation with appropriate patient selection and target definition, and where SBRT is becoming the standard of care [14,40,42]. Since the implementation of SBRT varies greatly between centers and the optimal dose and fractionation are not well defined, standardization of implementation and use is essential to enable relevant inter-institutional comparisons. Our review also highlights that well-conducted prospective clinical trials are required to define the role of SBRT in the curative treatment of newly diagnosed MCHN. One example could be the study NCT04435938.

## Figures and Tables

**Figure 1 cancers-16-02096-f001:**
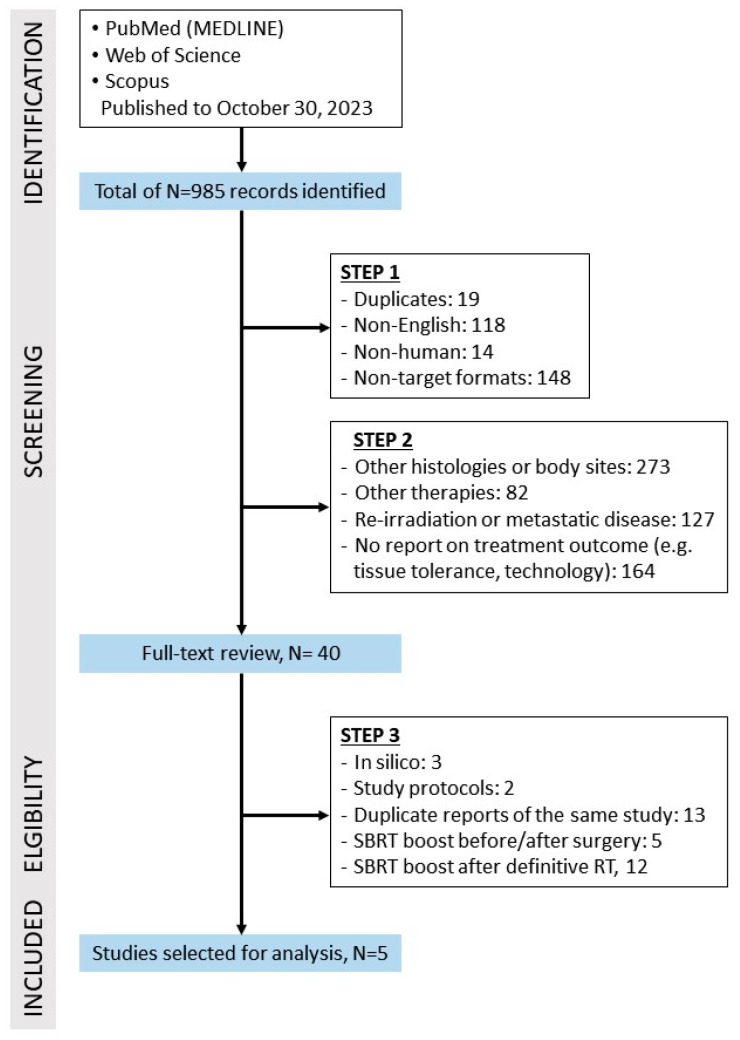
Study selection based on inclusion and exclusion criteria.

**Table 1 cancers-16-02096-t001:** PICOS inclusion criteria for literature screening.

Population	Patients with treatment-naive mucosal cancer of the head and neck.
Intervention	Stereotactic body radiotherapy, defined as the delivery of potentially ablative radiation doses, typically in 1–5 fractions (hypofractionation), with high precision to a well-defined and small target volume.
Control	No control groups.
Outcome	Primary outcomes: toxicity and local control. Secondary outcomes: regional control, progression-free survival, overall survival, quality of life.
Study design	Prospective and retrospective studies with more than 3 patients included.

**Table 2 cancers-16-02096-t002:** Summary of included studies in terms of patient and tumor characteristics.

Author (Ref)	Study Type	Recruitment Period	N	Age, Median (Years)	Criteria/Reasons for Referral to SBRT	Tumor Site	Histology	T-Stage	N-Stage	TNM Stage
Kawaguchi et al., 2012 [19]	R	3/2006–9/2007	14	73	to avoid surgery/hospitalization, to achieve good cosmetic/functional result	oral cavity = 11 maxillary sinus = 2 oropharynx = 1	SCC	T2 = 5 T3 = 3 T4 = 6	N0 = 13 N1 = 1	n.r.
Vargo et al., 2014 [20]	R	2002–2013	5/12	88	medical inoperability, well-lateralized tumor	base of tongue = 2 oral cavity = 1 maxillary sinus = 1 larynx = 1	SCC	T2 = 1 T4 = 4	N0 = all	II = 1 IVA = 4
Sher et al., 2019 [21]	P	11/2013–3/2017	12/29	61 ^1^	PS 0-1	glottis	SCC	Tis-T2	N0 = all	n.r.
Al Assaf et al., 2020 [22]	R	10/2011–10/2016	48/114	81 ^1^	comorbidities, poor PS	different sites, n.s.	SCC, 69.3% ^1^	n.s.	n.s.	n.r.
Gogineni et al., 2020 [23]	R	8/2011–6/2018	45/66	80 ^1^	advanced age, comorbidities, bulky disease	oral cavity, 23 larynx, 5 hypopharynx 4	SCC, 67% ^1^	n.r.	n.r.	n.r.

N—Number of patients; SBRT—Stereotactic body radiotherapy; R—Retrospective; P—Prospective; PS—Performance status; SCC—Squamous cell carcinoma; n.r.—Not reported; n.s.—Not specified. ^1^ Applies to the entire study group.

**Table 3 cancers-16-02096-t003:** The risk of bias assessment for included the studies using the ROBINS-I tool.

Author (Ref)	Pre-Intervention		At Intervention	Post-Intervention				Overall Judgement of Bias
	Bias Due to Confounding	Bias in Selection of Participants into the Study	Bias in Classification of Intervention	Bias Due to Deviation from Intended Intervention	Bias Due to Missing Data	Bias in Measurement of Outcomes	Bias in Selection of the Reported Results	
Kawaguchi et al., 2012 [19]	serious	serious	serious	serious	moderate	moderate	moderate	serious
Vargo et al., 2014 [20]	serious	serious	serious	moderate	moderate	low	moderate	serious
Sher et al., 2019 [21]	moderate	moderate	low	low	low	low	low	moderate
Al Assaf et al., 2020 [22]	serious	serious	serious	serious	serious	moderate	moderate	serious
Gogineni et al., 2020 [23]	serious	serious	serious	serious	serious	serious	moderate	serious

**Table 4 cancers-16-02096-t004:** Details on the practice of stereotactic body radiotherapy.

Author (Ref)	Delivery Platform and Immobilization	PTV, cm^3^	Margins	Elective Nodal RT	Dose Prescription	RT Schedule	BED_10_	EQD2_10_	Image Guidance	Systemic Therapy
Kawaguchi et al., 2012 [19]	CyberKnife, and custom-made mountpiece, thermoplastic mask	n.r.	n.r.	No	80–85% isodose line, covering periphery of GTV	35–42 Gy/5 fx	59.5–77.28 Gy	49.58–64.4 Gy	daily, 2 orthogonal kV x-ray sources	yes, all (S-1, 5-FU)
Vargo et al., 2014 [20]	CyberKnife Triology TrueBeam and thermoplastic mask	41	PTV = GTV; since 2012: PTV = GTV + 2–5 mm	No	n.r.	44 Gy/5 fx 3 fx/wk	82.72 Gy	68.93 Gy	daily X-Sight skull tracking or CBCT or ExacTrac	yes, 2 pts (CMb)
Sher et al., 2019 [21]	CyberKnife and five-point thermoplastic mask	5.3 ^1^	CTV = IGTV + 2 mm ^2^ PTV = CTV + 3 mm	No	57–90% isodose line (median, 86%) ^1^	42.5 Gy/5 fx; 2 fx/wk	78.63 Gy	65.52 Gy	n.r.	no
Al Assaf et al., 2020 [22]	Elekta Synergy (IMRT, VMAT) and five-point thermoplastic mask	GTV, 33.2 ^1^	PTV = GTV + 3–5 mm	12/114 pts: immediately adjacent nodal basins; PTV = CTV + 3–5 mm PTV D95% = 25 Gy	GTV D95	40–50 Gy/5 fx (89.8% of pts) ^1^; 1–2 fx/wk	72–100 Gy	60–83.33 Gy	daily, kV CBCT,	no
Gogineni et al., 2020 [23]	Gantry-based linear accelerator (VMAT) and standard thermoplastic mask	32 ^1^	PTV_primary_ = GTV + 2 mm PTV_nodal_ = CTV + 2 mm	NF primary: lateral retropharyngeal nodes; N+ pts: ipsilateral nodal regions II-IV; CTV = 30 Gy	PTV D95% = 98–100%	35 Gy/5 fx (50% of pts) ^1^ 40 Gy/5 fx (50% of ts); ^1^ 2 fx/wk	59.5–72 Gy	49.58–60 Gy	daily, CBCT or kV	yes, 48% ^1^

PTV—Planning target volume; RT—Radiotherapy; BED_10_—Biological effective dose (α/ß = 10 Gy); EQD2_10_—Equivalent dose in 2 Gy fractions (α/ß = 10 Gy); n.r.—not reported; GTV—Gross tumor volume; CTV—Clinical target volume; IGTV—Internal gross tumor volume; NF—Nasopharyngeal; N+—Node positive; pts—patients; fx—fraction(s); wk—week; CBCT—Cone beam computed tomography; kV—kilovaltage; 5-FU—5-fluorouracil; CMb—Cetuximab. ^1^ Applies to the entire study group. ^2^ Tumor < 2 mm of the anterior commissure, CTV = commissure and 2 mm of contralateral cord; tumor < 2 mm of arytenoid cartilage, CTV = entire arytenoid cT2 tumor, CTV = ipsilateral vocal cord and adjacent parapharyngeal space.

**Table 5 cancers-16-02096-t005:** Follow-up and toxicity data.

Author (Ref)	FUP, Median (mos)	LC	RC	Survival	Acute Toxicity G ≥ 3	Late Toxicity G ≥ 3	QoL
Kawaguchi et al. 2012 [19]	36	71.4% at 3 yrs ^3^	11/14 (78.6%)	OS 78.6% at 3 yrs	none	Osteoradionecrosis,1 (7.1%), ^4^	n.r.
Vargo et al., 2014 [20]	20	80%	100%	40%: DWOD 3 NED 2	G3 dysphagia,1 (8.3%) ^1^	G3 late mucositis, 1 (8.3%) ^1^	UW-QoL-R: improved/stable QoL compared to baseline in 71% of pts (N = 7) ^1^
Sher et al., 2019 [21]	25.7	100%	100%	100%	none	G3 soft tissue and arytenoid necrosis, 1 (8.3%)	MDADI: no decline after treatment VHI: rapid improvement at FUP
Al Assaf et al., 2020 [22]	mean, 10.5	78.9% at 2 yrs	LRC outside PTV 89% at 1 yr	PFS 23.7 mos OS 40% at 2 yrs ^2^	G3, 52.1% ^1^: - dermatitis, 32.5% - mucositis, 19.6%	G4 osteonecrosis, skin ulceration, 2 (4.2%)	n.r.
Gogineni et al., 2020 [23]	15	73% at 1 yr ^1^ (13 pts with 24 mos FUP = 69%)	73% at 1 yr ^1^	OS 64% at 1 yr ^1^	G3, 2 (3%) ^1^: - dysphagia, 1 (1.5%) - anorexia, 1 (1.5%)	none	n.r.

FUP—Follow-up; mos—Months; LC—Local control; RC—Regional control; G—Grade; QoL—Quality of life; yrs—years; LRC—Locoregional control; PTV—Planning target volume; OS—Overall survival; DWOD—Died without evidence of disease; NED—No evidence of disease; PFS—Progression-free survival; n.r.—Not reported; UW-QoL-R—University of Washington quality of life revised; N—Number of patients; MDADI—MD Anderson dysphagia inventory; VHI—Voice handicap index. ^1^ Applies to the entire study group. ^2^ Assessed from Figure 2A in ref. [19]. ^3^ Four of 14 patients had additional SBRT for recurrent lesion. ^4^ Six months after the second SBRT course due to local recurrence.

## Data Availability

The datasets analyzed during the current study are available from the corresponding author on reasonable request.

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
