# Peer review of "Stereotactic Body Radiotherapy as a Curative Treatment for De Novo Mucosal Carcinoma of the Head and Neck: A Feasible Alternative Option for Fragile Patients with Small Lesion: A Systematic Review†"

_cancers, 2024, doi:10.3390/cancers16112096_

Round 1

Reviewer 1 Report

Comments and Suggestions for Authors

Although the authors have done good work by compiling the data on the role of Stereotactic Body Radiotherapy as a Primary Curative Treatment for de novo Mucosa Head and Neck Cancer. However, the authors need to work further on manuscript to make it comprehensive.

1. More data(research results) needs to searched and included.

2. The manuscript should explicit methods to critically appraise and synthesize on the topic.

3.  Search all the latest relevant studies and include in the manuscript.

Comments on the Quality of English Language

Minor English and scientific language edit is needed.

Author Response

Reviewer #1

Although the authors have done good work by compiling the data on the role of Stereotactic Body Radiotherapy as a Primary Curative Treatment for de novo Mucosa Head and Neck Cancer. However, the authors need to work further on manuscript to make it comprehensive.

  1. More data(research results) needs to searched and included.

Response:

In fact, there are several articles in the literature on the subject of stereotactic radiotherapy in patients with head and neck tumors. Unfortunately, none of these articles fulfill the necessary criteria that we set out in the protocol of our study. Here are a few examples:

  • Siddiqui F, Patel M, Khan M, et al. Stereotactic body radiation therapy for primary, recurrent, and metastatic tumors in the head-andneck region. Int J Radiat Oncol Biol Phys 2009;74:1047–53. – only 2 patients with primary mucosal head and neck carcinoma included (in the primary group)
  • Kodani N, Yamazaki H, Tsubokura T, et al. Stereotactic body radiation therapy for head and neck tumor: disease control and morbidity outcomes. J Radiat Res 2011;52:24–31. – the radiation dose cannot be considered ablative (median D90 = 35 Gy, median number of fractions = 5)
  • Karam SD, Snider JW, Wang H, et al. Survival outcomes of patients treated with hypofractionated stereotactic body radiation therapy for parotid gland tumors: a retrospective analysis. Front Oncol 2012;2:55. – no mucosal head and neck tumors treated (only parotid gland tumors)
  • Amini A, McDermott JD, Gan G, et al. Stereotactic body radiotherapy as primary therapy for head and neck cancer in the elderly or patients with poor performance. Front Oncol 2014;4:274. –3 cases described; only one being mucosal head and neck carcinoma
  • Schwartz DL, Sosa A, Chun SG, et al. SBRT for early stage glottic larynx cancer-initial clinical outcomes from a phase I clinical trial. PLoS One 2017;12:e0172055. – preliminary report; final results were reported in Sher DJ et al (Int J Radiat Oncol Biol Phys 2019) which is included in the present systematic report

In several other reports, stereotactic body radiotherapy was used as a boost after previous conventional fractionated irradiation, for example:

  • Chen HHW, Tsai S-T, Wang M-S, et al. Experience in fractionated stereotactic body radiation therapy boost for newly diagnosed nasopharyngeal carcinoma. Int J Radiat Oncol Biol Phys 2006;66:1408–14.
  • Lee DS, Kim YS, Cheon JS, et al. Long-term outcome and toxicity of hypofractionated stereotactic body radiotherapy as a boost treatment for head and neck cancer: the importance of boost volume assessment. Radiat Oncol 2012;7:85.

  1. The manuscript should explicit methods to critically appraise and synthesize on the topic.

Response:

We have added a more detailed description of the type of data extracted from the selected studies and an explanation of the risk of bias assessment to the Methods section (page 4, paragraph 2):

“The following data were extracted from the eligible studies: type of the study (retrospective, prospective case series, randomized clinical trial), number and age of patients, tumor location, histology and stage, SBRT treatment details (delivery platform, planning target volume [PTV] size, elective nodal irradiation, dose prescription data, biological effective dose [BED], EQD2, image guidance), use of systemic therapy and toxicity details (acute, late grade 3 or higher), treatment outcome (local and regional control), patient survival (progression-free survival and overall survival) and quality of life assessments results. The risk of bias of the included studies was assessed independently by 2 of 3 authors (PS, MK, GP) using the ROBINS-I tool for non-randomized studies, which classifies studies into those at low, moderate, serious and critical risk of bias [17]. Disagreements were discussed until consensus was reached.

Added reference (17):

Sterne JA, Hernán MA, Reeves BC, Savović J, Berkman ND, Viswanathan M, et al. ROBINS-I: a tool for assessing risk of bias in non-randomised studies of interventions. Br Med J (Clin Res Ed) 2016; 355:i4919.doi: 10/12/2016;355.

  1. Search all the latest relevant studies and include in the manuscript.

Response:

The present systematic review included published full articles up to and including October 2023. In an additional search using the same search terms in the PubMed (MEDLINE), Web of Science and Scopus databases for the period from November 1, 2023 to April 28, 2024, we found no newer articles that met the stated criteria: SBRT as a primary curative treatment, mucosal carcinoma of the head and neck, potentially ablative RT dose, stereotactic technique.

Comments on the Quality of English Language

Minor English and scientific language edit is needed.

Response:

The text was additionally reviewed by an English-language co-author of the manuscript, who made necessary corrections.

Reviewer 2 Report

Comments and Suggestions for Authors

This systematic review included 5 studies, mostly retrospective. As the author described there are many limitations to reviewing this treatment. However, this review provided a good summary of this treatment modality at this time.

Author Response

Reviewer #2

This systematic review included 5 studies, mostly retrospective. As the author described there are many limitations to reviewing this treatment. However, this review provided a good summary of this treatment modality at this time.

Response:

The authors welcome the positive criticism of the text: indeed, the literature on stereotactic radiotherapy for de novo mucosal head and neck carcinoma with curative intent is sparse and very heterogeneous.

Reviewer 3 Report

Comments and Suggestions for Authors

The authors conducted a systematic review of SBRT for de novo head and neck cancer, which is an area with few publications.  This make systematic review challenging and data gathering is mainly descriptive. There are a few retrospective studies that the authors did not include.  It is also unclear why the authors stop literature search by 2020 and not include later published work.  The authors shall highlight the potential indication for such approach, limitation of such approach, and also limitation of this analysis.

Comments on the Quality of English Language

The manuscript's quality of English language is excellent.

Author Response

Reviewer #3

The authors conducted a systematic review of SBRT for de novo head and neck cancer, which is an area with few publications. This make systematic review challenging and data gathering is mainly descriptive. There are a few retrospective studies that the authors did not include.  It is also unclear why the authors stop literature search by 2020 and not include later published work. The authors shall highlight the potential indication for such approach, limitation of such approach, and also limitation of this analysis.

Response:

The present systematic review included published full articles up to and including October 2023. In an additional search using the same search terms in the PubMed (MEDLINE), Web of Science and Scopus databases for the period from November 1, 2023 to April 28, 2024, we found no newer articles that met the stated criteria: primary curative treatment, mucosal carcinoma of the head and neck, potentially ablative RT dose, stereotactic technique.

In fact, there are several articles in the literature on the subject of stereotactic radiotherapy in patients with head and neck tumors. Unfortunately, none of these articles fulfill the required inclusion criteria listed above. Here are a few examples:

  • Siddiqui F, Patel M, Khan M, et al. Stereotactic body radiation therapy for primary, recurrent, and metastatic tumors in the head-andneck region. Int J Radiat Oncol Biol Phys 2009;74:1047–53. – only 2 patients with primary mucosal head and neck carcinoma included (in the primary group)
  • Kodani N, Yamazaki H, Tsubokura T, et al. Stereotactic body radiation therapy for head and neck tumor: disease control and morbidity outcomes. J Radiat Res 2011;52:24–31. – the radiation dose cannot be considered ablative (median D90 = 35 Gy, median number of fractions = 5)
  • Karam SD, Snider JW, Wang H, et al. Survival outcomes of patients treated with hypofractionated stereotactic body radiation therapy for parotid gland tumors: a retrospective analysis. Front Oncol 2012;2:55. – no mucosal head and neck tumors treated (only parotid gland tumors)
  • Amini A, McDermott JD, Gan G, et al. Stereotactic body radiotherapy as primary therapy for head and neck cancer in the elderly or patients with poor performance. Front Oncol 2014;4:274. –3 cases described; only one being mucosal head and neck carcinoma
  • Schwartz DL, Sosa A, Chun SG, et al. SBRT for early stage glottic larynx cancer-initial clinical outcomes from a phase I clinical trial. PLoS One 2017;12:e0172055. – preliminary report; final results were reported in Sher DJ et al (Int J Radiat Oncol Biol Phys 2019) which is included in the present systematic report

In several other reports, stereotactic body radiotherapy was used as a boost after previous conventional fractionated irradiation, for example:

  • Chen HHW, Tsai S-T, Wang M-S, et al. Experience in fractionated stereotactic body radiation therapy boost for newly diagnosed nasopharyngeal carcinoma. Int J Radiat Oncol Biol Phys 2006;66:1408–14.
  • Lee DS, Kim YS, Cheon JS, et al. Long-term outcome and toxicity of hypofractionated stereotactic body radiotherapy as a boost treatment for head and neck cancer: the importance of boost volume assessment. Radiat Oncol 2012;7:85.

Reviewer 4 Report

Comments and Suggestions for Authors

From a biostats and clinical epidemiology point of view, here are some comments for the Authors:

- I do suggest to rename the manuscript as a narrative review, due to at least two main reasons:

1) 4 retrospective case series and 1 prospective phase I study can not be truly labeled as the result of a systematic review

2) you only partially complied with Cochrane methodology principles (i.e. risk of bias, GRADE, publication bias and so on)

- caption for figure 1 is not adequate (Figure 1. This is a figure. Schemes follow the same formatting)

- tables 2-3-4 , a vast heterogeneity affects all the aspects of the selected studies, moreover a wide lackness of TNM info is another really critical point

-  duration of the mean/median follow-up, only the median/IQR has to be reported

- being a narrative review, all its limits have to be clearly stated in the discussion

Author Response

Reviewer #4

From a biostats and clinical epidemiology point of view, here are some comments for the Authors:

- I do suggest to rename the manuscript as a narrative review, due to at least two main reasons:

1) 4 retrospective case series and 1 prospective phase I study cannot be truly labeled as the result of a systematic review

2) you only partially complied with Cochrane methodology principles (i.e. risk of bias, GRADE, publication bias and so on)

Response:

The authors are of the opinion that the term "systematic review” aptly describes the study with regard to the methodology used and the content covered. The small number of included articles and/or patients that fulfill the predefined selection criteria cannot be a reason not to call the review "systematic".

The Methods section (page 4, paragraph 2) has been expanded to include an explanation of the assessment of the risk of bias. The results are described in the Results section (3.1. Characteristics of the eligible studies) and presented in Table 2.

  • Methods (page 4, paragraph 2):

“The risk of bias of the included studies was assessed independently by 2 of 3 authors (PS, MK, GP) using the ROBINS-I tool for non-randomized studies, which classifies studies into those at low, moderate, serious and critical risk of bias [17]. Disagreements were discussed until consensus was reached.”

Added reference (17):

Sterne J, Herna´n M, Reeves B, Savović J, Berkman ND, Viswanathan M, et al. ROBINS-I: a tool for assessing risk of bias in non-randomised studies of interventions. Br Med J (Clin Res Ed) 2016; 355:i4919.doi: 10/12/2016;355.

  • Results (page 5, 3.1. Characteristics of the eligible studies)

“According to the ROBINS I tool for assessing the risk of bias in the included studies, the overall risk of bias was rated as severe for all four retrospective studies and moderate for the prospective study. The risk of bias in each domain is summarized in Table 2. As the retrospective studies are case series, an important source of bias can be found in all three domains.”

Table 2. The risk of bias assessment for included the studies using the ROBINS-I tool.

Author (Ref)

Pre-intevention

At intervention

Post-intervention

Overall judgement

of bias

Bias due to confounding

Bias in selection of participants into the study

Bias in classification of intervention

Bias due to deviation from intended intervention

Bias due to missing data

Bias in measurement of outcomes

Bias in selection of the reported results

Kawaguchi et al, 2012 (18)

serious

serious

serious

serious

moderate

moderate

moderate

serious

Vargo et al, 2014 (19)

serious

serious

serious

moderate

moderate

low

moderate

serious

Sher et al, 2019 (20)

moderate

moderate

low

low

low

low

low

moderate

Al Assaf et al, 2020 (21)

serious

serious

serious

serious

serious

moderate

moderate

serious

Gogineni et al, 2020 (21)

serious

serious

serious

serious

serious

serious

moderate

serious

- caption for figure 1 is not adequate (Figure 1. This is a figure. Schemes follow the same formatting)

Response:

We apologize for the error in transcribing the text. The title of Figure 1 reads: "Study selection based on inclusion and exclusion criteria".

- tables 2-3-4, a vast heterogeneity affects all the aspects of the selected studies, moreover a wide lackness of TNM info is another really critical point

Response:

We are aware of the heterogeneity of the studies and the resulting limitations. We have modified/inserted the following text in the last paragraph of the Discussion:

“Our review is limited by the caveats of each of the studies analyzed, ranging from the small number of cases included to incomplete data reporting and the heterogeneity of the studies analyzed. Not only are all but one study retrospective and the overall risk of bias was considered severe: even within individual studies, it can be challenging to clearly separate those with de novo MCHN from other patients. A significant limitation resulting from the retrospective nature of most studies is also the lack of information on the TNM stage of the disease, which would allow a fair assessment and comparison of local control and survival outcomes between studies.

-  duration of the mean/median follow-up, only the median/IQR has to be reported

Response:

Table 5 shows the follow-up time as indicated in the individual articles: as median in 4 articles and as mean value in one article. The interquartile range (IQR) is only given in one article.

- being a narrative review, all its limits have to be clearly stated in the discussion

Response:

The limitations of our review are listed (and supplemented as indicated above) in the last paragraph of the Discussion section.

Round 2

Reviewer 4 Report

Comments and Suggestions for Authors

line 204, risk of bias is reported in table 2 and not 3

Author Response

Reviewer #4

Comments and Suggestions for Authors: line 204, risk of bias is reported in table 2 and not 3

Response:

In fact, the table is incorrectly marked in the text (it is Table 2 and not Table 3), which we have corrected (marked in green) and we apologize for the error.

No further comments or questions were received from the Editor and reviewers #1, #2 and #3 in the second round of review.
